# Adaptation to Hypoxia May Promote Therapeutic Resistance to Androgen Receptor Inhibition in Triple-Negative Breast Cancer

**DOI:** 10.3390/ijms23168844

**Published:** 2022-08-09

**Authors:** Nikita Jinna, Padmashree Rida, Max Smart, Mark LaBarge, Tijana Jovanovic-Talisman, Rama Natarajan, Victoria Seewaldt

**Affiliations:** 1Department of Population Science, City of Hope Comprehensive Cancer Center, Duarte, CA 91010, USA; 2Rowland Hall, Salt Lake City, UT 84102, USA; 3Department of Molecular Medicine, City of Hope Comprehensive Cancer Center, Duarte, CA 91010, USA; 4Department of Diabetes Complications and Metabolism, City of Hope Comprehensive Cancer Center, Duarte, CA 91010, USA

**Keywords:** hypoxia, hypoxia-inducible factors, adaptation, androgen receptor, triple-negative breast cancer, therapeutic resistance

## Abstract

Triple-negative breast cancer (TNBC) surpasses other BC subtypes as the most challenging to treat due to its lack of traditional BC biomarkers. Nearly 30% of TNBC patients express the androgen receptor (AR), and the blockade of androgen production and AR signaling have been the cornerstones of therapies for AR-positive TNBC. However, the majority of women are resistant to AR-targeted therapy, which is a major impediment to improving outcomes for the AR-positive TNBC subpopulation. The hypoxia signaling cascade is frequently activated in the tumor microenvironment in response to low oxygen levels; activation of the hypoxia signaling cascade allows tumors to survive despite hypoxia-mediated interference with cellular metabolism. The activation of hypoxia signaling networks in TNBC promotes resistance to most anticancer drugs including AR inhibitors. The activation of hypoxia network signaling occurs more frequently in TNBC compared to other BC subtypes. Herein, we examine the (1) interplay between hypoxia signaling networks and AR and (2) whether hypoxia and hypoxic stress adaptive pathways promote the emergence of resistance to therapies that target AR. We also pose the well-supported question, “Can the efficacy of androgen-/AR-targeted treatments be enhanced by co-targeting hypoxia?” By critically examining the evidence and the complex entwinement of these two oncogenic pathways, we argue that the simultaneous targeting of androgen biosynthesis/AR signaling and hypoxia may enhance the sensitivity of AR-positive TNBCs to AR-targeted treatments, derail the emergence of therapy resistance, and improve patient outcomes.

## 1. Introduction

Triple-negative breast cancer (TNBC) continues to be the most difficult-to-treat BC subtype. TNBCs do not express the conventional BC targets currently exploited for therapeutic intervention, including the estrogen receptor (ER), progesterone receptor (PR), and human epidermal growth factor receptor 2 (HER2) [1]. TNBCs are inherently more aggressive, with an increased risk of death within 5 years post-diagnosis and a higher rate of distant metastasis and recurrence following surgery compared to other breast cancer subtypes [2,3]. Precision medicine approaches to targeting TNBC are complicated by inter- and intra-tumoral heterogeneity.

Over the past few decades, the androgen receptor (AR) has been highlighted as an alternate endocrine target of interest for the subset of TNBC patients positive for AR expression. AR expression is assessed by immunohistochemistry (IHC); about 20%–40% of TNBC tumors express AR [4]. AR is most highly expressed in the “luminal AR (LAR)” molecular subtype of TNBC; however, in vitro and in vivo experiments involving AR-inhibitors and AR knockdown have demonstrated that non-LAR molecular subtypes of TNBC may also be critically dependent on AR protein for viability, proliferation, migration, and invasion [5]. These results led to the notion that AR-targeted therapies could exploit this AR dependence and benefit multiple molecular subtypes of AR-positive TNBC. Although several AR inhibitors (such as bicalutamide, abiraterone acetate, and enzalutamide) have elicited significant antitumor activity in advanced clinical trials with AR-positive TNBC patients, approximately only one in five experienced sustained clinical benefit. This limited success suggests the development of therapeutic resistance among the majority of patients [4,5,6,7,8]. However, the mechanisms of resistance to AR inhibition in TNBC remain underexplored. Moving the needle on our understanding of this resistance to AR-targeted therapies is imperative for improving therapeutic efficacy and for the advancement of AR inhibitory drugs to the FDA-approval stage.

Hypoxia, or limited oxygen availability due to insufficient blood supply, is a unique feature of most solid tumors [9]. Rapid and uncontrolled tumor growth can lead to tumors outgrowing their surrounding vasculature, which leads to a significant drop in normal oxygen levels (from 9%–2%) and the development of hypoxia (<2%). Oxygen is necessary for tumor cell metabolism and proliferation. Thus, slow-dividing cells located in hypoxic areas of tumors can escape most anticancer agents that kill rapidly dividing cells [10]. Furthermore, hypoxia induces profound gene expression changes that promote tumor cells’ survival, the growth of new vasculature, energy metabolism, lineage plasticity, and heterogeneity, often conferring an aggressive and metastatic phenotype [11,12]. Thus, hypoxia has been significantly implicated in therapeutic resistance to multiple anticancer agents and underlies poor patient prognosis [13,14,15].

The adaptive response to hypoxia is regulated by hypoxia-inducible factors (HIFs), which modulate hypoxic gene expression and regulate oxygen homeostasis. Heterodimeric HIF is composed of an oxygen-labile α subunit and a stable β subunit; HIF α/β heterodimers are thus comprised of one of three α subunits (HIF-1α, HIF-2α, or HIF-3α) and one of two β subunits (HIF-β and aryl hydrocarbon receptor nuclear translocator or ARNT). The HIF-α subunit undergoes proteasomal degradation under normoxic conditions; in contrast, HIF-β and ARNT are constitutively expressed and reside in the nucleus. Under hypoxic conditions, stabilized HIF-α translocates to the nucleus, where the active HIF-α/ARNT or HIF-α/HIF-β complex activates the transcription of a large group of target genes after binding to cognate hypoxia-responsive elements (HREs) [16,17,18].

Accumulating evidence shows that hypoxia can hinder the success of androgen deprivation therapy (ADT) and AR inhibitors in AR growth-driven prostate cancers by increasing AR expression and/or activity and upregulating other compensatory signaling pathways to promote castration-resistant prostate cancer (CRPC) [19,20,21,22,23,24]. Dual therapeutic targeting of the AR and hypoxia signaling axes has been shown to circumvent this hypoxia-mediated drug interference and delay the onset of CRPC [25,26]. However, the role of hypoxia in potentially mediating therapeutic resistance to AR inhibitors in TNBC remains unknown. HIF-1α levels have been shown to be elevated in TNBCs relative to other BC subtypes demonstrating that TNBCs are able to grow under hypoxic conditions and thus resist treatments [27]. Herein, we explore and encourage investigation into the potential role of hypoxia in the resistance to therapies that target androgen or AR signaling in TNBC and provide evidence that the dual targeting of the hypoxia and AR signaling axes may help avert AR-targeted therapy resistance in AR-positive TNBC patients.

## 2. Tried and Tested: Androgen Synthesis and AR Signaling as Therapeutic Targets for TNBC

Since AR-driven/-dependent TNBC is a potentially actionable subtype, androgen- and AR-targeting therapies have been explored for their ability to stymie disease progression in AR-positive TNBC [28]. Clinical trials involving TNBC patients with both early-stage and advanced disease have evaluated the clinical benefit rates (CBRs) for patients under treatment with anti-androgen drugs, including bicalutamide or enzalutamide, which were administered as single agents as well as in combination therapies. Drugs that reduce androgen production are also being evaluated for AR-positive TNBC and have shown varying degrees of efficacy but were largely supportive of targeting the androgen/AR axis in AR-positive TNBC.

Bicalutamide is FDA-approved for the treatment of advanced PC in combination with a luteinizing hormone-releasing hormone analog. Bicalutamide is orally available and is a competitive inhibitor of AR that permits AR nuclear localization and binding to chromatin but preferentially recruits AR corepressors rather than coactivators [29]. A clinical study conducted by investigators at the Memorial Sloan Kettering Cancer Center (NCT00468715/TBCRC011) showed that daily treatment with 150 mg bicalutamide led to 19% CBR lasting longer than 6 months and a median progression-free survival (PFS) of 12 weeks in ER-negative/PR-negative/AR-positive metastatic BC patients [6]. These proof-of-concept data in AR-positive BC provided an impetus to pursue next-generation AR antagonists because bicalutamide can have partial agonist effects [30]. PC patients who develop resistance to bicalutamide are oftentimes responsive to the next-generation antagonist enzalutamide, which (a) has an eightfold higher affinity for AR than bicalutamide and (b) is a robust inhibitor of AR signaling that blocks AR nuclear translocation and coactivator interactions and attenuates the DNA binding capacity of AR [31].

In a phase I dose-escalation study in patients with metastatic BC in 2013, enzalutamide was well tolerated at 160 mg daily. In a 2015 phase II study, among the AR-positive patients evaluable for response, the CBR (defined as complete response + partial response + stable disease) at 16 weeks was 42% with patients receiving 160 mg enzalutamide daily. About 34% of the participants in the study continued to show clinical benefits after 24 weeks. Furthermore, when a novel, binary, gene signature-based predictive biomarker that was reflective of AR signaling was used to stratify patients, the outcomes for diagnostic-positive patients were more favorable in all response measures, including CBR16 (39% vs 11%), CBR24 (36% vs 6%), and median PFS (16.1 weeks vs 8.1 weeks), when compared with those of patients who tested negative for this companion diagnostic. Interestingly, this companion diagnostic was a stronger predictor of response to enzalutamide than AR expression (evaluated by IHC) alone. These responses to an AR antagonist supported (a) the development of anti-AR therapy for the treatment of a subgroup of patients with TNBC [8], (b) the development of companion diagnostics that better identify the androgen/AR-dependence of tumors, and (c) the pre-planned incorporation of such predictive biomarkers (that are superior to AR positivity alone) and novel biomarker discovery into clinical trials. The ENDEAR trial, which was scheduled to be an international, double-blind, randomized, placebo-controlled, three-arm phase III study in TNBC patients who tested positive for the companion diagnostic developed during the phase II trial previously mentioned, was unfortunately abandoned in 2017 based on a joint decision by the partnering pharmaceutical companies, citing a need to improve the clarity on the role of androgen signaling in BC. Based on the promising data available regarding the metastatic setting, a phase II study is currently evaluating the feasibility of enzalutamide as endocrine therapy in the adjuvant setting (1 year of enzalutamide 160 mg orally daily) for patients with early-stage (Stages I–III) TNBC of the LAR subtype (NCT02750358). This single-arm trial has met its primary endpoint of feasibility; however, the median overall survival (OS) has not yet been reached. Apalutamide and darolutamide are two promising new-generation AR inhibitors under evaluation in phase III clinical trials in patients with non-metastatic CRPC (NCT01946204 and NCT02200614, respectively). Apalutamide, structurally similar to enzalutamide, has exhibited a similar success rate as enzalutamide at a lower dosage in an LNCaP xenograft mouse model but does not induce AR nuclear translocation or DNA binding [32]. Darolutamide is particularly interesting owing to its ability to also antagonize the AR mutants F876 L, W741 L, and T877A [33]. The applicability and efficacy of these drugs in the context of AR-positive TNBC are yet to be extensively studied.

Agents that target steroidogenic enzymes to impair intracrine and adrenal androgen biosynthesis and concomitantly reduce downstream estrogen synthesis have also been tested in clinical trials for TNBC; abiraterone acetate, a CYP17 inhibitor, is one such therapeutic, which is a robust, orally available, selective inhibitor of both 17α-hydroxylase and C17, 20-lyase. In a phase II multicenter trial that aimed to assess the safety and efficacy of abiraterone acetate in combination with prednisone (which has to be included to offset the increase in aldosterone production that results from reduced cortisol levels) in women with metastatic or inoperable locally advanced AR-positive TNBC, the CBR at 6 months was 20.0% and median PFS was 2.8 months. This study suggested that some TNBC patients with molecular apocrine-like tumors may benefit from the coadministration of abiraterone acetate and prednisone. Seviteronel, an oral, selective CYP17-lyase inhibitor and AR antagonist, is also currently being evaluated as a novel option for the treatment of AR-positive TNBC because this agent does not cause a decrease in cortisol and no steroid supplementation is required. Preliminary pharmacokinetic data from a large phase I/II trial for seviteronel in AR-positive TNBC and ER-positive metastatic BC suggested that the bioavailability of seviteronel may be sex-specific [34]. The phase II trial in this population established the promising CYP17-lyase inhibitory activity of seviteronel, as evidenced by an early and potent reduction in both estradiol and testosterone [35]. Pre-clinical data also advocate for the administration of seviteronel as a radiotherapy-sensitizing agent in AR-positive TNBC [36].

Since other oncogenic aberrations may co-exist with AR dependency, combination regimens involving AR inhibitors and other targeted therapies are also underway for AR-positive TNBC. Palbociclib, an orally administered and highly selective and reversible CDK4/6 inhibitor, prevents the onset of the S phase, resulting in cytotoxicity and growth arrest. Studies have shown that luminal phenotypes, elevated cyclin D1 and Rb protein expression, and reduced p16 expression are all associated with sensitivity to Palbociclib [37]. AR-positive TNBC often exhibits a luminal gene expression profile and has intact Rb protein; these observations provide a rationale for combining AR inhibitors with Palbociclib for the treatment of AR-positive TNBC. A phase I/II trial (NCT02605486) in patients with metastatic BC [35] met its pre-specified endpoint of PFS of at least 6 months. A trial testing the combination of bicalutamide and ribociclib (another CDK4/6 inhibitor) is also underway (NCT03090165). AR antagonists have also been paired with drugs that target the PI3K/AKT/mTOR pathways because AR-positive BCs are often associated with activating PIK3CA mutations and pAKT [38,39,40]. Pre-clinical studies showed that LAR cell lines are significantly more sensitive to NVP-BEZ235 (a dual PI3 K/mTOR inhibitor) when compared to cell lines of basal-like TNBC molecular subtypes [40]. These findings catalyzed an investigator-initiated, multi-institutional phase I/II study (TBCRC032) that evaluated the safety and efficacy of enzalutamide alone or in combination with the PI3K inhibitor, taselisib, in patients with metastatic AR-positive (≥10%) BC [41]. The combination of enzalutamide and taselisib significantly increased the CBR in the AR-positive TNBC patients. Importantly, analyses confirmed earlier findings that AR protein expression alone is insufficient for identifying patients with AR-dependent tumors, and a greater understanding of TNBC molecular subtypes and AR splice variants may identify patients more or less likely to respond to AR antagonists. Since the LAR subtype is generally resistant to conventional multidrug neoadjuvant chemotherapy (NAC) regimens and exhibits low rates of pathologic complete response, or pCR [42], a phase II clinical trial was carried out to assess whether combining AR inhibition with NAC would improve pCR or near-pCR in early-stage AR-positive (10%) TNBC patients treated with enzalutamide and weekly paclitaxel (NCT02689427). The results of this trial showed that 33.3% of TNBC patients who did not respond to conventional NAC showed a pCR with the enzalutamide and paclitaxel combination; notably, all responders showed an upregulated androgen response pathway as measured by transcriptomic analysis in pre-treatment biopsies [43].

## 3. The Arch Nemesis: Hypoxia as a Barrier in Androgen/AR Signaling Inhibition in Cancer

A significant body of evidence shows that tumor hypoxia interferes with therapies that disrupt AR signaling or reduce androgen levels in PC. A recent study showed that enzalutamide induces hypoxia and microenvironment adaptation and that two cytokines—interleukin-8 (IL8) and vascular endothelial growth factor A (VEGF-A)—upregulated by this treatment-induced hypoxia may contribute to this treatment resistance [44]. This study also showed that the concurrent inhibition of both IL8 and VEGF-A in PC pre-clinical models prolonged tumor sensitivity to enzalutamide. One primary mechanism of this hypoxia-mediated resistance to AR-targeted therapies is via upregulating AR signaling. Several studies have shown that overexpressing HIF-1α in PC cells stimulated AR signaling with the androgen dihydrotestosterone (DHT), enhanced AR transcriptional activity, and subsequently increased secretion of VEGF [21,22,24,25,26,27,28,29,30,31,32,33,34,35,36,37,38,39,40,41,42,43,44,45,46]. Specifically, hypoxia increased AR translocation to the nucleus and recruitment to the prostate-specific antigen (PSA) promoter [19,21]. It was also shown via chromatin immunoprecipitation that HIF-1α interacts with AR on the human PSA gene promoter in DHT-stimulated PC cells, suggesting that HIF-1α and AR may be cooperating to activate AR target genes [47]. Collectively, these findings suggest that hypoxia enhances the AR signal transduction pathway. As a result, hypoxia has been reported to drive insensitivity to androgen/AR-targeted therapies in PC. In PC cells, DHT-mediated AR signaling induced HIF-1α expression, and the dual treatment of enzalutamide and HIF-1α inhibition abrogated cell growth, induced apoptosis, and reduced VEGF levels more effectively than the administration of each treatment alone [25]. These studies thus provide a strong impetus for combining therapeutic targeting of the HIF-1α and AR pathways as a strategy to combat enzalutamide resistance.

It was also discovered that HIF-1α coordinates AR translocation to the nucleus via binding to β-catenin and AR to form a ternary complex that binds to the androgen response elements [48]. Under hypoxic conditions, the dietary polyphenol, resveratrol, reduced HIF-1α levels and inhibited β-catenin nuclear accumulation to suppress hypoxia-induced AR transactivation in prostate tumors possibly in a proteasome-independent manner [26]. These findings suggest that targeting the Wnt/β-catenin signaling axis in combination with androgen/AR signaling may also circumvent hypoxia-mediated resistance to AR-targeted therapies. Furthermore, the AR-signaling-mediated induction of HIF-1 was shown to occur via the phosphatidylinositol 3-kinase (PI3K) signaling as the inhibition of this pathway blocked HIF-1 activation [46]. Lastly, it was recently shown that hypoxia simultaneously upregulates HIF-1α and transient receptor potential melastatin subfamily member 7 (TRPM7) in androgen-independent PC cells and that knockdown of TRPM7 inhibited hypoxia-induced migration and invasion via the increased RACK1-mediated degradation of HIF-1α [49].

Some groups have begun to unravel other mechanisms of intra-tumoral hypoxia-mediated resistance to androgen/AR signaling inhibition. Hypoxia-mediated resistance to the potent second-generation AR inhibitor, enzalutamide, was shown to mechanistically occur in PC via restoration of glucose-6-phosphate isomerase (GPI), which is transcriptionally repressed by AR in hypoxia and maintains glucose metabolism and energy homeostasis under hypoxic conditions [22]. AR inhibition restores GPI, which was shown to promote resistance to this inhibition via redirecting glucose flux from the androgen/AR-dependent pentose phosphate pathway to the hypoxia-induced glycolysis pathway. Thus, GPI inhibition was shown to circumvent enzalutamide resistance in vivo. Cancer stem cells have also been implicated in treatment resistance to AR-targeted therapies in PC [50,51]. ADT has been shown to increase cancer stem cell numbers in prostate tumors and HIF signaling in response to hypoxia and induce the expression of stemness and EMT genes that promote the emergence of these cancer stem cells [52,53,54]. Furthermore, HIF-1α was shown to regulate CD44, which is expressed on stem-like BC cells, by increasing the presence of CD44 molecules and the percentage of CD44 positive cells under hypoxic conditions [55]. Thus, the expansion of the cancer stem cell subpopulation upon the induction of adaptive HIF signaling may also be conferring resistance to AR-targeted therapies.

## 4. Double Trouble: Hypoxia in Triple-Negative Breast Cancer

Hypoxic regions have been observed in over 50% of breast tumors but more frequently in TNBC. Genomic profiling revealed high HIF-1α signaling activity in basal-like/TNBCs [56]. Specifically, HIF-1α expression was observed to be highly upregulated in TNBCs to withstand a hypoxic tumor microenvironment. TNBCs have also been shown to upregulate bicarbonate transporters (NDBTs) such as *SLCA4* and *SLC4A5*, which regulate tumor pH levels when adapting to hypoxic conditions [57].

The enhanced hypoxic signaling observed in TNBC has been suggested to underlie advanced progression and treatment resistance in TNBC, which may also be undermining the efficacy of AR inhibition, as seen in Figure 1. Specifically, HIF-1α upregulates Snail expression, which increases the migration and invasiveness of BC cells by downregulating E-cadherin. Farnesyltransferase treatment of TNBC cells to inhibit HIF-1α expression decreased mRNA levels of HIF-1α pathway genes such as Snail, glucose transporter 1, pyruvate dehydrogenase kinase 1, and lactate dehydrogenase A [58]. HIF-1α and HIF-2α silencing in TNBC xenografts significantly reduced tumor growth [59]. In TNBC, HIF-1α upregulated complement 1 q binding protein (C1QBP), which blocked metastasis of TNBC cells and increased their sensitivity to paclitaxel [60]. The depletion of C1QBP in TNBC cells also decreased VCAM-1 expression and suppressed the activation of hypoxia-induced protein kinase C-nuclear factor-κappa B signaling [60]. The nano-treatment of TNBC in vitro and in vivo with the hypoxia-targeting drug tirapazamine, effectively abrogated tumor cell growth and progression in hypoxic regions [27]. The inhibition of TNBC hypoxia-induced NDBTs was shown to notably suppress migration and invasion in vitro and in vivo via attenuating the NDBT-mediated hypoxic phospho-signaling activation and modulating the expression of critical EMT-related genes, such as vimentin, which prevents downregulation of E-cadherin [57]. The inhibition of the hypoxia-induced carbonic anhydrase IX in TNBC cell lines impaired their ability to form new vasculature and mammospheres as well as metastasize [61]. Hence, targeting hypoxia in TNBC has been shown to increase tumor sensitivity to chemotherapies such as cisplatin, doxorubicin, and 5-fluorouracil [62,63,64,65].

Hypoxia has also been shown to predict prognosis in TNBC. Positive HIF-1α IHC expression significantly correlated with greater tumor size, higher histological grade, positive lymph node status, and higher tumor TNM stage as well as poorer postoperative survival [66]. The enrichment of a hypoxia-related three-gene signature model was shown to predict poorer overall survival (OS) in TNBC patients [67]. Furthermore, a high combined hypoxia and immune-base gene signature score predicted a poorer prognosis among TNBC patients [68].

## 5. How Hypoxia May Be Culpable in Resistance to Androgen/AR-Targeting Therapies in TNBC: Potential Ways to Divide and Conquer

### 5.1. AR-Targeting Therapies May Be Causing the Loss of an Ally, ER-β

Although AR therapy is a promising prospect for TNBC, it has so far consistently proven beneficial for only a subset of the participants in the clinical trials. Consequently, researchers have been attempting to identify predictive biomarkers that may indicate a high likelihood of benefit from androgen/AR-targeting therapies. Full-length ER-β protein has been observed in 50%–90% of ER-α-negative BCs [69,70]. Evidence shows that AR upregulates ER-β expression in ER-α-negative BCs [71] and ER-β expression correlates with improved disease-free survival and good prognosis in TNBC [72]. More importantly, it has been shown that ER-β inhibits the transcriptional activity of HIF-1α by inducing ubiquitination and degradation of ARNT leading to the reduction of active HIF-1a/ARNT complexes [73]. It has therefore been suggested that combining AR-targeting treatments—especially the inhibitors of intracellular AR signaling—with selective agonists of ER-β (for patients whose TNBC is ER-β-positive) may stimulate the anti-proliferative activity of ER-β while reducing the HIF-1α-dependent hypoxic response that could undermine the efficacy of the therapy. Thus, ER-β may not only bring significant value as a predictive biomarker but may also substantially enhance the efficacy of androgen/AR-targeted treatments [74]. Furthermore, future clinical trials evaluating only AR-targeting agents in TNBC should ideally focus on BRCA-wild, AR-positive, and ER-β-negative patients.

### 5.2. Upregulation of Compensatory Pathways Involving the Glucocorticoid Receptor and Wnt/β-Catenin- and PI3K/AKT/mTOR-Mediated Signaling

Enzalutamide-resistant tumors have also been shown to upregulate glucocorticoid receptors (GR), which may drive the expression of some AR-regulated genes and thereby decrease dependence on AR [75]. Ligand-dependent activation of GR was shown to enhance hypoxia-dependent gene expression and HRE activity [76]. The upregulation of GR has been suggested to occur via the binding of HIF-1α to one or more sites at the GR promoter to enhance GR transcription [77]. Thus, targeting hypoxia along with AR in AR-enriched TNBCs may also circumvent AR-targeted therapy resistance. Furthermore, Wnt/β-catenin signaling has been reported to be one of the most significantly deregulated pathways in CRPC and is considered a compensatory pathway that PCs activate in response to ADT [67,78,79,80,81]. Lee and colleagues showed that AR growth-driven PCs are likely low in Wnt/β-catenin signaling due to the increased affinity of β-catenin for AR rather than TCF4, as the suppression of AR activity via androgen deprivation in PC cells increased β-catenin/TCF4 target gene transcription [82]. DHT stimulation of ER-negative BC cells resulted in the downregulation of the TCF/LEF family genes likely because of the previously described mechanism in PC [83]. As mentioned prior, HIF-1α forms a ternary complex with AR and β-catenin to facilitate AR nuclear translocation to upregulate AR target genes. Thus, targeting the hypoxia signaling axis in addition to targeting oncogenic AR signaling may be more effective in averting AR treatment resistance in AR-positive TNBC.

As previously discussed, Lehmann and colleagues discovered that AR-positive TNBCs display a greater frequency of PI3KCA and AKT1 mutations [40,84]. AR phosphorylation by phosphorylated AKT was shown to inhibit AR-induced apoptosis, resulting in increased cell survival [85]. The activation of the PI3K/AKT pathway has been implicated in the resistance to AR-targeted therapies in PC [86]. Particularly, PI3K pathway inhibition was shown to increase the transcriptional activity of AR [87]. Thus, the concomitant inhibition of AR and the PI3K pathway in clinical trials with PC patients has been lackluster [88,89]. As mentioned previously, AR signaling induces hypoxic signaling via upregulating the PI3K/AKT pathway. Hence, targeting the hypoxia signaling axis along with AR signaling in AR-positive TNBC may represent a more effective treatment strategy. The PI3K/AKT pathway can also activate mTOR and phosphorylated mTOR has been associated with positive AR expression in TNBC [90,91]. Thus, hypoxia inhibition may also circumvent dysregulation in mTOR signaling in AR-positive TNBCs.

### 5.3. Compensatory Upregulation of Steroid Hormone Transporters

Studies aimed at gaining insight into the basis for the resistance to abiraterone treatment in PC have revealed that the SLCO1B3 gene (which encodes the OATP1B3 steroid hormone transporter that regulates intra-tumoral androgen uptake and concentration) is induced by abiraterone [92]. This effect is mediated by three miRNAs including hsa-miR-579-3p, which binds to the 3′ UTR of SLCO1B3 [92]. Previous studies have demonstrated that SLCO1B3 is also induced by hypoxia [93]. Increases in OATP1B3 expression result in enhanced androgen uptake and faster disease progression in PC. The downregulation of hsa-miR-579-3p under hypoxic conditions or by treatment with abiraterone increases the levels of OATP1B3 in PC cells and drives abiraterone resistance. Whether the treatment of AR-positive TNBCs with abiraterone acetate also leads to increases in the androgen uptake transporter OATP1B3 via a similar mechanism to drive therapy resistance, especially under hypoxic conditions merits urgent investigation.

### 5.4. Upregulation of Pathways That Underlie Tumor Expansion and Metastases in TNBCs

Targeting the hypoxia signaling axis in AR-positive TNBC may simultaneously target other mechanisms of resistance induced upon AR signaling inhibition such as angiogenesis, epithelial-mesenchymal transition (EMT), epigenetic reprogramming, and AR-independent lineage plasticity [89]. This upregulation may be facilitated by HIF-1α, which upregulates genes involved in these processes such as VEGF, Snail, Axl, the enhancer of zeste homologue 2 (EZH2), and CD24 [94,95,96,97,98]. Thus, therapeutically targeting hypoxia could inhibit multiple mechanisms underlying the resistance to AR-targeted therapies in TNBC. The inhibition of the histone acetylation reader bromodomain and extraterminal (BET) was shown to overcome enzalutamide resistance in CRPC as well as decrease tumor growth and vascularization of TNBC xenografts under hypoxic conditions [99,100].

### 5.5. Production of Constitutively Active AR Splice Variants and Hyper-Stable AR Transcripts

In the context of PC, the induction of a variety of AR splice variants (AR-Vs) that occurs as an acute response to selection pressures, such as castration, or exposure to therapeutic mainstays, such as enzalutamide and abiraterone, is well documented [101,102,103,104,105,106]. AR-Vs emerge as a result of alternative splicing and/or structural rearrangements in the AR gene. Studies have uncovered that biologically active AR-Vs may require full-length AR (AR-FL) to activate the endogenous AR target genes [107]. The most common AR-V is AR-V7, which is nuclear-localized and contains exons 1, 2, 3, and a cryptic exon 3b but lacks the ligand-binding domain (LBD), which is the docking site for drugs such as enzalutamide; the absence of the LBD in this variant receptor makes it a constitutively active, gain-of-function, ligand-independent transactivator that maintains AR signaling even in an androgen-depleted environment [103,105]. Studies in 22Rv1 PC cells have shown that HIF-1α can heterodimerize with AR-V7 (and AR-FL), facilitate AR-V7′s nuclear localization, and promote resistance to enzalutamide. Multiple clinical studies have linked AR-V7 expression in PC cells to the resistance to AR antagonists, such as enzalutamide, and disease progression, and non-invasive assays to detect this splice variant in circulating tumor cells may indicate whether individual PC patients are likely to respond to these drugs or progress quickly [108,109]. A recent study showed that (a) miR-361-3p, a tumor suppressor whose expression is decreased in multiple cancer types including PC, normally binds to the 3′UTRs of AR-V7 (which differs from the 3′ UTR of AR-FL), and the MAP kinase-interacting serine/threonine kinase 2 (MKNK2) transcript, leading to suppression of AR-V7 and MKNK2 protein expression and enzalutamide sensitivity in vitro and in vivo and (b) enzalutamide decreases the expression of miR-361-3p by upregulating hypoxic signaling [110]. Another study involving PC cell lines with acquired resistance to AR antagonists showed that the levels of AR mRNA transcript and protein were markedly increased following AR-antagonist treatment [111]. Notably, this study found that 20% of AR transcripts had a 3 kb deletion within the normally 6.7 kb long 3′ UTR and that this shorter AR UTR splice variant, which was also detectable in patients’ post-treatment tumor samples and PC patient sera, had a significantly increased stability/half-life, which potentially conferred a survival advantage to the tumor cells. Yet another recent study found that an oncogenic lncRNA KDM4A-AS1, which notably increased in CRPC cell lines and cancer tissues and was associated with unfavorable outcomes, promoted the stability of the USP14-AR-FL/AR-Vs complex and the de-ubiquitination of AR/AR-Vs. By repressing proteolysis of AR-FL/AR-Vs, the lncRNA KDM4A-AS1 enhanced CRPC drug resistance to enzalutamide [112]. Importantly, KDM4A-AS1 is a hypoxia-responsive gene and is known to be transactivated by HIF-1α. Furthermore, in hepatocellular carcinoma, hypoxia-induced KDM4A-AS1 increased HIF-1α expression by activating the AKT pathway to form a positive feedback loop [113]. The proteasome-associated deubiquitinating enzyme USP14 also binds and stabilizes AR in AR-positive BC cells including TNBC [114]; the highly likely possibility that the hypoxia-induced lncRNA KDM4A-AS1 binds and stimulates the de-ubiquitination of AR in TNBC also warrants investigation. In sum, these studies lend strong support to the idea that the synergistic inhibition of AR and hypoxia pathways may be necessary to reduce the stability of AR-Vs and enhance the efficacy of androgen-/AR-targeted treatments in PC. Several AR-Vs, including AR-V7, have been detected in TNBC cell lines and breast tumors [115,116,117]. AR-V7 was upregulated by enzalutamide in primary ER-α-negative breast tumors and functional studies further confirmed the growth-stimulating activity of AR-V7 in an ER-α-negative BC context. Furthermore, unlike in PC cells, AR-V7 activated a transcriptome distinct from AR-FL in BC cells, suggesting contextual specificity. Taken together, these findings suggested that AR-Vs, including AR-V7, may be clinically significant in TNBC patients receiving AR-targeted treatments [116]. It is now recognized that alternative splicing reflects a “readiness” for the adaptation to environmental changes and that hypoxia-driven alternative splicing is a potent force powering tumor pathogenesis and progression in diverse cancer types [118]. Hypoxia influences spliceosome assembly, expression levels/activities/localization of splicing factors, and miRNA synthesis and maturation, spurring adaptive changes in the proportions of alternatively spliced transcripts; in fact, the astonishingly high frequency and reach of this phenomenon had led to its classification as the “11th hallmark” of cancer, which in turn, impacts all other hallmarks of cancer and is a critical determinant of therapeutic resistance. Therefore, moving forward, the co-suppression of hypoxia-driven alternative splicing alongside AR-targeting therapy could be a clinical imperative for AR-positive TNBC.

### 5.6. Induction of Centrosome Amplification, Chromosomal Instability, and Intra-Tumoral Heterogeneity

Another reason it may be beneficial to target hypoxic signaling in AR-positive TNBC is to thwart the induction of the centrosome amplification (CA) that is known to result from hypoxia [119,120]. The pan-cancer occurrence of CA has led to the designation of this phenotype as a hallmark of cancer. Tumor microenvironmental hypoxia induces the HIF-1α-dependent overexpression of PLK4 [121], AURORA A [122], and CYCLIN D [120], resulting in CA in a wide range of cancer types. Amplified centrosomes drive chromosomal instability (CIN), spur an increase in intra-tumoral heterogeneity [123], and augment the metastatic potential of cancer cells in aneuploidy-independent ways [124]. A prognostic gene expression-based score CA20, which is based on the expression of 20 CA-associated genes (AURKA, CCNA2, CCND1, CCNE2, CDK1, CEP63, CEP152, E2F1, E2F2, LMO4, MDM2, MYCN, NDRG1, NEK2, PIN1, PLK1, PLK4, SASS6, STIL, and TUBG1), portended poorer outcomes in BC patients in the METABRIC and TCGA datasets [125]. When tested across 15 different cancer types from TCGA, the CA20 signature was positively associated with the 99-metagene hypoxia signature in multiple cancers [126], supporting the notion that hypoxic cancers exhibit higher CA and consequently, greater CIN. Nakada et al. showed that the upregulation of the HIF-1α transcriptional target miR-210 due to hypoxia induced widespread CA and CIN in renal cell carcinoma [127]. TNBC cells exhibit high frequency and severity of CA [128], which contributes to the failure of a variety of therapies by promoting CIN, invasiveness, and intra-tumoral heterogeneity. Since cancer cells, unlike healthy somatic cells, are endowed with supernumerary centrosomes, they rely on centrosome clustering mechanisms to coalesce their excessive centrosome complement into two polar groups to achieve pseudo-bipolar mitosis and evade the catastrophic sequelae of multipolar mitoses. As a result, centrosome declustering drugs comprise a highly promising class of nontoxic, cancer-cell-selective therapeutic agents [129] that ought to be used in any scenario wherein hypoxic conditions are likely to occur including in combination with AR-targeting therapies for TNBC.

## 6. Fresh Air: Perspectives Regarding the Future of AR Therapy in Hypoxic TNBC

AR has long been acknowledged as an alternative and actionable endocrine target for AR-positive, ER-negative BCs such as TNBC. Unlike the other BC endocrine targets, ER and PR, the therapeutic inhibition of the AR pathway has been less successful and the reasons for this therapeutic failure are more complex. Although the concept of targeting the androgen/AR axis is not inherently flawed, the data clearly show that we need to better anticipate that hypoxia is likely to diminish the clinical efficacy of these strategies and design customized therapies that address these resistance mechanisms preemptively. Accumulating evidence shows that TNBCs are more enriched in hypoxic signaling than the other BC subtypes and that hypoxia may undergird the increased resistance to anticancer drugs, such as endocrine therapies and monoclonal antibodies against HER2, which is frequently observed in TNBC [130,131,132,133]. Furthermore, cross-talk between the AR and HIF pathways has been extensively documented as has the adaptive role of hypoxia in promoting the emergence of AR point mutations, “androgen-indifferent” AR-Vs, and CA that further enable tumor cells to bypass the effects of AR-targeted treatments. Thus, it is critical to investigate the potential role of hypoxia in mediating AR inhibition resistance in AR-dependent TNBCs.

The afore-mentioned studies provide a sobering reminder that AR inhibition of tumors can lead to the activation or upregulation of adaptive pathways, including the PI3K/AKT, mTOR, GR, and Wnt/β-catenin pathways, which are themselves oncogenic. Although targeting these pathways in combination with AR inhibitors such as enzalutamide has elicited some preclinical and clinical success, the inhibition of these pathways is often met with challenges such as the activation of other compensatory pathways. For example, the inhibition of the PI3K/AKT/mTOR signaling network has led to the upregulation of the interconnected pathways RAS-MEK-ERK and JAK2/STAT5 signaling [134]. The GR pathway has also been known to exhibit cross-talk with other signaling networks such as MAPK, PI3K/AKT, JAK/STAT, and NOTCH [135,136,137,138]. Targeting the Wnt/β-catenin pathway has been met with a high risk for off-target effects and cytotoxicities because the signaling pathway plays a critical role in stem cell maintenance and exhibits cross-talks with other major signaling pathways [139,140] Furthermore, the inhibition of these signaling networks may neglect the importance of the successful inhibition of other mechanisms of AR therapy resistance such as EMT, stem cell expansion, and angiogenesis.

Inhibiting hypoxia in combination with AR therapy in AR-positive TNBCs may represent a more effective multipronged approach to circumventing the multiple mechanisms of AR therapy resistance. As mentioned prior, the hypoxia signaling network is interconnected with the compensatory pathways of AR inhibition such as PI3K/AKT and Wnt/β-catenin. Thus, targeting the hypoxia signaling axis may concomitantly target or interfere with the downstream effects of these and other compensatory pathways. Additionally, hypoxic signaling promotes the development of other mechanisms of AR therapy resistance such as increasing alternative splicing, EMT, stem cell regeneration, and angiogenesis. Thus, therapeutically targeting hypoxia may also suppress these additional mechanisms.

Therapies aimed at inhibiting the hypoxic signaling axis have primarily been designed to directly or indirectly block HIF-1α and/or HIF-2α expression or function. Specifically, these HIF inhibitors interfere with HIFα mRNA expression, protein synthesis, dimerization, DNA binding, and transcriptional activity, as well as protein stabilization and accumulation [140]. These agents remain in the early stages of clinical trials primarily due to a lack of understanding of the hypoxic signaling axis, deteriorating diffusion geometry in hypoxic tissue areas that interfere with drug delivery, and a lack of appropriate patient selection.

Therefore, moving forward, we envisage that the thrust of our efforts would most likely need to be in the following directions: we need to increase pre-clinical and clinical investigations into (a) the AR signaling axis and its rich “conversations” with the tumor microenvironment in TNBC, (b) the molecular mechanisms by which hypoxia engenders and/or exacerbates androgen-/AR-independence and therapy resistance in AR-positive TNBC, (c) the effects of combining AR and HIF inhibitors in pre-clinical models that more accurately recapitulate the hypoxic conditions within solid tumor microenvironments, (d) AR-variant protein expression in TNBC tumor specimens, particularly post-treatment with AR-targeted therapies, alone and in combination with HIF inhibitors, and (e) the prognostic value of HIF-1α, or hypoxia gene signatures as biomarkers could improve risk-stratification for AR-positive TNBC patients as well as patient segmentation for HIF inhibitor treatment. A pre-clinical investigation should include assessing the response of AR-enriched TNBC cell lines and patient-derived xenograft models to dual treatment with current-generation AR and HIF-1α inhibitors compared to each inhibitor administered alone. Above all, and given the fact that AR IHC alone is a poor proxy for AR dependence and is a woefully inadequate patient pre-selection criterion and predictor of response to AR-targeted treatments, we need to focus efforts on identifying better tumor-intrinsic biomarkers that integrate information about true AR dependence and the tumor’s TNBC molecular subtype, that take into consideration and are attuned to the hypoxia status of the tumor, the AR splice variant status, the AR mutational and ER-β status, and the CA landscape of the tumor. Novel biomarker discovery also needs to be pre-planned and built into the design of clinical trials. The identification of superior, multi-omic predictive biomarkers will enable the thoughtful design of better, tumor-informed clinical trials involving carefully selected patient subsets, within which specific treatment combinations may deliver the greatest and most durable clinical benefits and in the end be practice-changing. Since angiogenesis, induced upon hypoxia, is a major culprit of AR therapy resistance, it may be beneficial to evaluate angiogenesis markers and incorporate anti-angiogenic therapies in the clinic.

Angiogenesis is often the culprit in hypoxia-induced therapy resistance; it is no wonder then that several targeted anti-angiogenics have been FDA-approved for epithelial malignancies. Anti-angiogenics are credited with decreasing tumor metastasis and promoting the clinical efficacy of chemotherapy, immunotherapy, and radiation therapy [141]. It is well established that the blood vessels that grow into and nourish tumors are far from normal; tumor vessels tend to be highly fenestrated, abnormally permeable, and excessively sinuous, and the distance between the vessels and tumor cells is typically heterogeneous [142,143,144,145]. Moreover, instead of being strictly unidirectional and consistent, the blood flow within the tumor vessels may be irregular, stagnant, and even retrograde. As a result, interstitial hypertension and intercellular matrix edema are oft-found features of solid tumors [143]. Importantly, studies have shown that in regions where the tumor vasculature is abnormal, the interstitial tissue is hypoxic; thus, hypoxia may serve as a surrogate for abnormal vasculature. Such abnormal vasculature, which results from an imbalance in the delicate balance between proangiogenic and anti-angiogenic factors in the tumor microenvironment, has deep clinical implications because it impairs the delivery of therapeutics to solid tumors. Indeed, evidence shows that anti-angiogenics are effective, at least in part, because they transiently “re-normalize” tumor vasculature and alleviate interstitial hypoxia [143,144,145].

To facilitate the individualized treatment of tumors with anti-angiogenics, agents and probes that allow direct, non-invasive, and dynamic in vivo visualization of hypoxic regions of breast tumors, can inform clinical decision making by providing real-time information regarding the hypoxia status of the tumor, the extent of the re-normalization of the vessel architecture, as well as the changes in the interstitial delivery of therapeutics [146]. A 2016 study provided proof of principle that 18F-fluoromisonidazole-PET ([18F]-FMISO-PET) binds specifically to hypoxic regions and serves as a read-out for hypoxia reversion and is a tracer for visualizing both the re-normalization of tumor vasculature and vessel functionality, as well as the changes in the delivery of chemotherapeutics to different patient-derived xenografts [147]. However, there are certain practical drawbacks of [18F]-FMISO-PET such as a rather sluggish rate of clearance from healthy tissues as well as from the blood, a low target-to-background ratio, a relatively short half-life, and the need for patient imaging about 3 h after tracer administration. Second-generation hypoxia radiotracers, such as [18F]-fluoroazomycin-arabinoside ([18F]F-FAZA), which has greater hydrophilicity compared with [18F]F-FMISO, afford improved target-to-background ratios and superior pharmacokinetic profiles and are under active development [148]. It is therefore reasonable to envision that the efficacy of anti-AR/androgen therapies may be enhanced by the co-administration of anti-angiogenics that may re-sensitize AR-positive TNBCs to these AR-targeting therapies. The concomitant use of biomarkers such as [18F]-FMISO-PET or [18F]F-FAZA may galvanize the clinical implementation of combination regimens involving anti-AR treatments as well as anti-angiogenics/anti-hypoxia therapeutics because they enable a more optimal choice and dynamic monitoring of personalized treatments, as well as response assessment during therapy; indeed, such a strategic combination of diagnostics with therapeutics—termed theragnostics—embraces a level of precision oncology that pursues nimble shifts in therapy tailored based on changes in the molecular characteristics of tumors. Thus, the precise identification and quantitation of hypoxic levels via hypoxia-specific in vivo imaging techniques, such as PET and Single Photon Emission Computed Tomography (SPECT), will be critical to successfully designing optimal therapeutic strategies to co-target hypoxia along with AR in each individual TNBC patient [39,146,149].

The establishment of AR as a robust risk-prognostic and therapeutic target for AR-positive TNBC patients still requires a deeper understanding of the AR signaling axis and the interplay between AR signaling and the TNBC tumor microenvironment. Our discussion aims to encourage further investigation into an important aspect of the tumor microenvironment, hypoxia, and its potential role in undermining current AR-targeted therapies in AR-enriched TNBC. We assert that the increased understanding of the interaction between the AR and hypoxia signaling axes in TNBC could renew interest in exploiting AR for the clinical management of TNBC patients and lead to breakthroughs in AR therapy administration.

## Figures and Tables

**Figure 1 ijms-23-08844-f001:**
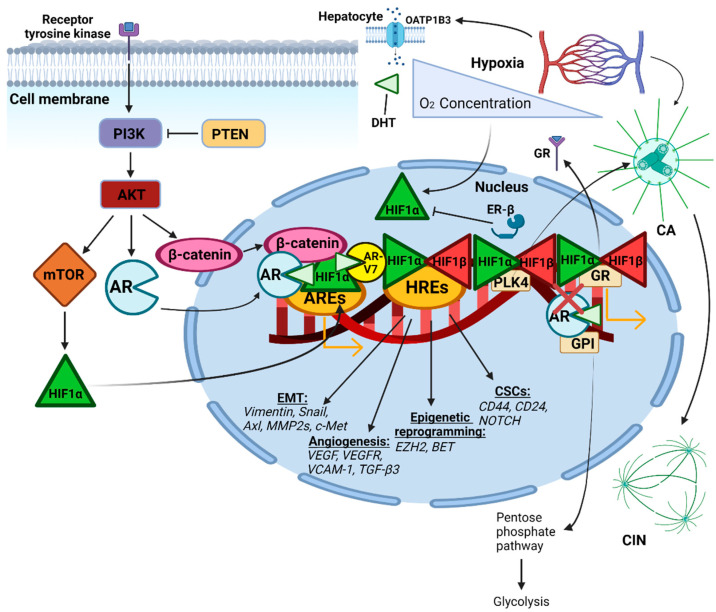
Illustration of the potential role of hypoxia in mediating resistance to AR therapy in TNBC. Adaptation of AR-positive TNBC cells to hypoxic conditions may be interfering with the efficacy of current AR-targeted therapeutics via (i) downregulating ER-β to upregulate HIF-1α transcription, (ii) upregulating compensatory pathways that enhance HIF-1α transcriptional activity, such as GR, Wnt/β-catenin, and PI3K/AKT/mTOR signaling, (iii) restoration of AR-repressed GPI, (iv) upregulation of steroid hormone receptors such as OATP1B3 upon hypoxia induction to increase hepatocytic androgen uptake, (v) HIF-1α-mediated induction of genes facilitating EMT, angiogenesis, epigenetic reprogramming, and cancer stem cell renewal, (vi) increased HIF-1α heterodimerization with AR-V7 to alternatively upregulate AR signaling, and (vii) HIF-1α-mediated induction of CA and subsequent CIN. Abbreviations: OATP1B3, organic anion-transporting polypeptide 1B3; DHT, dihydrotestosterone; PI3K, phosphoinositide 3-kinases; PTEN, phosphatase and tensin homolog; AKT, protein kinase B; mTOR, mammalian target of rapamycin; AR, androgen receptor; AR-V7, androgen receptor splice variant 7; ER-β, estrogen receptor beta; HIF-1α, hypoxia-inducible factor 1 subunit alpha; HIF-1β, hypoxia-inducible factor 1 subunit beta; ARE, androgen receptor element; HRE, hypoxia response element; GR, glucocorticoid receptor; GPI, glucose-6-phosphate isomerase; CA, centrosome amplification; CIN, chromosome instability; MMP2, matrix metallopeptidase 2; c-MET, tyrosine-protein kinase MET; VEGFR, vascular endothelial growth factor receptor; VCAM-1, vascular cell adhesion protein 1; TGF-β3, transforming growth factor beta-3; EZH2, enhancer of zeste homolog 2; bromodomain and extraterminal (BET); CD44, cell adhesion receptor 44.

## Data Availability

Not applicable.

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
