# Peer review of "Adaptation to Hypoxia May Promote Therapeutic Resistance to Androgen Receptor Inhibition in Triple-Negative Breast Cancer"

_ijms, 2022, doi:10.3390/ijms23168844_

Round 1

Reviewer 1 Report

This review covers the literature regarding the role of hypoxic tumour environment on effectiveness of AR inhibitors on TNBC. The authors postulate that simultaneous targeting of AR signaling and hypoxia may enhance the sensitivity of AR positive TNBC to AR-targeting treatments. It is a well written manuscript that covers all the relevant literature. I have the following comments:

1.       What search strategies were used for this review?

2.       Which databases were searched?

3.       Is this a systematic review?

4.       Could the authors explain in greater detail how would they set up a clinical (or pre-clinical) trial investigating the combination of AR inhibitors and HIF inhibitors?

Author Response

Reviewer 1:

This review covers the literature regarding the role of hypoxic tumour environment on effectiveness of AR inhibitors on TNBC. The authors postulate that simultaneous targeting of AR signaling and hypoxia may enhance the sensitivity of AR positive TNBC to AR-targeting treatments. It is a well written manuscript that covers all the relevant literature. I have the following comments:

Response: We are grateful for you positive and insightful comments. Please find our point-by-point responses to your questions and suggestion below:

  1. What search strategies were used for this review?
  2. Response: Great question. We performed pub med and google scholar searches to find supportive material for our article.
  3. Which databases were searched?
  4. Response: We utilized the pub med and google scholar databases to find the supportive articles.
  5. Is this a systematic review?
  6. Response: Great question. We would consider our review to be a critical review because we examine an array of relevant literature supporting a broad hypothesis that co-targeting hypoxia and AR in triple negative breast cancer could circumvent therapeutic resistance to AR inhibitors.
  7. Could the authors explain in greater detail how would they set up a clinical (or pre-clinical) trial investigating the combination of AR inhibitors and HIF inhibitors?
  8. Response: Excellent suggestion. We have further elaborated on how to pre-clinically investigate the effects of combining AR and HIF inhibitors in AR-positive TNBC in the discussion section in the second to last paragraph in the discussion section.

Reviewer 2 Report

This is a timely and well-written review on androgen-targeting therapeutic strategies for hypoxic triple negative breast cancers. I have a few suggestions to further improve the manuscript:

(1)            For an easier reading of Section 2 (Androgen synthesis and AR signaling as therapeutic targets for TNBC) I would recommend a summative table for all clinical studies and trials mentioned here.

(2)            As studies show that neither HIF-1alpha expression nor angiogenic markers present significant differences between triple negative breast cancer and breast cancers of other molecular subtypes (Yehia L, et al. PLoS One 2015), the authors should discuss some more personalised approaches for hypoxia identification / quantification. In view of this, the paper would benefit from a section on in vivo imaging of hypoxia in order to identify the hypoxic subgroups within resistant tumours. Hypoxia-specific imaging techniques via PET and SPECT further enable quantification of hypoxia allowing for a more personalised treatment. See for instance some of the latest publications:

https://pubmed.ncbi.nlm.nih.gov/35492341/

(3)            Also, since angiogenesis is often the culprit for hypoxia-induced resistance, angiogenic factors are potential therapeutic targets in breast cancer. Anti-angiogenic therapy should be therefore discussed in the paper. Potential reference: https://pubmed.ncbi.nlm.nih.gov/26778791/

Author Response

Reviewer 2:

This is a timely and well-written review on androgen-targeting therapeutic strategies for hypoxic triple negative breast cancers. I have a few suggestions to further improve the manuscript:

Response: Thank you for your kind feedback. Please find our point-by-point responses to your suggestions below:

  1. For an easier reading of Section 2 (Androgen synthesis and AR signaling as therapeutic targets for TNBC) I would recommend a summative table for all clinical studies and trials mentioned here. 
  2. Response: Great suggestion. We were initially going to create this table. However, after a recent literature search, we found multiple recent publications with a similar table, with up-to-date information on ongoing clinical trials in TNBC, already published. So, in order to avoid redundancy, so we did not include such a table. Instead, we have now included these references in section 2 for readers to refer to. However, if you still feel this table is necessary to include, we are happy to create one.
  1. https://www.nature.com/articles/s41523-020-00190-9
  2. https://pubmed.ncbi.nlm.nih.gov/27816190/

  1. As studies show that neither HIF-1alpha expression nor angiogenic markers present significant differences between triple negative breast cancer and breast cancers of other molecular subtypes (Yehia L, et al. PLoS One 2015), the authors should discuss some more personalised approaches for hypoxia identification / quantification. In view of this, the paper would benefit from a section on in vivo imaging of hypoxia in order to identify the hypoxic subgroups within resistant tumours. Hypoxia-specific imaging techniques via PET and SPECT further enable quantification of hypoxia allowing for a more personalised treatment. See for instance some of the latest publications: https://pubmed.ncbi.nlm.nih.gov/35492341/

Response: Excellent suggestion. We have included a paragraph addressing these points above the last paragraph in the discussion section.

  1. Also, since angiogenesis is often the culprit for hypoxia-induced resistance, angiogenic factors are potential therapeutic targets in breast cancer. Anti-angiogenic therapy should be therefore discussed in the paper. Potential reference: https://pubmed.ncbi.nlm.nih.gov/26778791/ 
  2. Response: Important suggestion. We have addressed this point in the third-to-last paragraph in the discussion section.